# The INVISION Talar Component in Revision Total Ankle Arthroplasty: Analysis of Early Outcomes

**DOI:** 10.3390/diagnostics14151612

**Published:** 2024-07-26

**Authors:** Bruno Valan, Albert T. Anastasio, Billy Kim, Alexandra Krez, Kevin A. Wu, Grayson M. Talaski, James Nunley, James K. DeOrio, Mark E. Easley, Samuel B. Adams

**Affiliations:** 1Division of Foot and Ankle, Department of Orthopaedic Surgery, Duke University School of Medicine, Durham, NC 27710, USA; bruno.valan@duke.edu (B.V.); albert.anastasio@duke.edu (A.T.A.); alexandra.krez@duke.edu (A.K.); james.nunley@duke.edu (J.N.); james.deorio@duke.edu (J.K.D.); mark.e.easley@duke.edu (M.E.E.); samuel.adams@duke.edu (S.B.A.); 2Department of Orthopaedic Surgery, Hospital for Special Surgery, New York City, NY 10021, USA; kimbil@hss.edu; 3Department of Orthopedics and Rehabilitation, University of Iowa, Iowa City, IA 52242, USA

**Keywords:** clinical outcome, INVISION, revision ankle arthroplasty, ankle joint, complication, innovation, joint reconstruction

## Abstract

Introduction: Launched in 2018 for revision total ankle arthroplasty (rTAA), the INVISION talar component addresses subsidence when poor talar bone stock is present. Due to the recency of the market-availability of the INVISION, studies evaluating its efficacy are lacking. This study presents the first analysis of early-term outcomes of patients undergoing rTAA with the INVISION talar component. Methods: This was a single-center, retrospective review of 28 patients undergoing rTAA with the INVISION talar component and INBONE II tibial component performed between 2018 and 2022. Data on preoperative characteristics, postoperative complications, secondary procedures, and survivorship were collected. The primary outcome measures were rates of major complications, re-operation, and implant failure. Secondary outcomes included post-operative changes in varus and valgus alignment of the tibia and talus. Results: The most common secondary procedures performed with rTAA were medial malleolus fixation (*n* = 22, 78.6%) and gastrocnemius recession (*n* = 14, 50%). Overall, 10.7% (*n* = 3) of patients underwent reoperation and 14.3% (*n* = 4) suffered major complications. Incidence of implant failure was 10.7% (*n* = 3). All reoperations were caused by infection. Mean varus alignment of the tibia and talus improved from 4.07 degrees and 4.83 degrees to 1.67 degrees and 1.23 degrees, respectively. Mean valgus alignment of the tibia and talus improved from 3.67 degrees and 4.22 degrees to 2.00 degrees and 2.32 degrees, respectively. Conclusions: In a series of 28 patients undergoing rTAA with the INVISION talar component, we discovered comparatively low rates of reoperation, major complication, and implant failure (10.7%, 14.3%, and 10.7%). The INVISION system appears to have a reasonable safety profile, but further studies evaluating long-term outcomes are required to assess the efficacy of the INVISION system.

## 1. Introduction

Severe ankle osteoarthritis (OA) remains a disabling disease that significantly impairs quality of life [1,2]. Over the past three decades, advancements in implant design and surgical technique have led to an increase in the employment of total ankle arthroplasty (TAA) as a treatment for end-stage OA of the ankle [1,3,4,5,6,7]. A recent MarketScan database analysis noted that in the United States (US), annual TAA volumes increased by 136.1% between 2009 and 2019 [2]. This trend is expected to be accompanied by a corresponding rise in the need for corrective procedures following failed TAA, necessitating further study of revision arthroplasty [4,5].

Prior studies on revision TAA (rTAA) have demonstrated superior outcomes when compared with other salvage procedures such as ankle arthrodesis for the treatment of the failed total ankle [4,8,9,10,11,12,13,14,15]. Egglestone et al. reported significant improvement in total Ankle Osteoarthritis Score (AOS), Pain Visual Analogue Score (VAS), and average Manchester–Oxford Foot Questionnaire (MOxFQ) in revision arthroplasty over arthrodesis [8]. Pfahl et al. showed lower failure rates, increased short-term survival, and better clinical outcomes with revision arthroplasty over ankle arthrodesis [9]. A recent randomized controlled trial conducted by Goldberg et al. comparing TAA and arthrodesis demonstrated improved MOXFQ-W/S scores in TAA patients, although this difference was not clinically significant [16]. While these findings may support the use of joint-preserving surgery, other studies provide evidence to the contrary.

A recent meta-analysis conducted by Li et al. compared patients’ clinical function scores and complications between those who received TAA and arthrodesis. The authors found no significant different in post operative Short Form-36 scores (MD = −1.19, 95% CI: −3.89 to 1.50, *p* = 0.39) or complication rates between the TAA and arthrodesis groups (RR = 0.95, 95% CI: 0.59 to 1.54, *p* = 0.85); however, patients who received arthrodesis were found to have improved post-operative Foot and Ankle Ability Measure scores compared to the TAA group (MD = 8.30, 95% CI: 1.01–15.60, *p* = 0.03) [17].

The decision between TAA and arthrodesis must also balance the patient factors such as age. Historically, young age and high physical demand were considered contraindications to TAA; however, some studies argue that patients under 50 years of age show better clinical results and higher satisfaction with TAA compared to ankle arthrodesis, despite a higher revision rate, due to its ability to preserve joint function and biomechanics, thus preventing degeneration of adjacent joints [18,19]. This ambiguity regarding the choice between ankle arthrodesis and TAA demonstrates a clear need for further work characterizing the differences in outcomes between fusion and arthroplasty. Moreover, there is scant literature evaluating the efficacy of specific implant systems in rTAA [7,15,20,21]. As more implant systems become available, discussion regarding device selection is increasingly relevant.

The INVISION total ankle replacement (Wright Medical/Stryker, Kalamazoo, MI, USA) was the first TAA system designed specifically with revision in mind. The INVISION system marked an important innovation in rTAA implant design by including a talar component, which can be used interchangeably with the INFINITY (Wright Medical/Stryker, Kalamazoo, MI, USA) and INBONE (Wright Medical/Stryker, Kalamazoo, MI, USA) TAA systems. The INVISION talar component includes a talar plate with two thickness options: a broad anatomic footprint with variable sizing in the sagittal plane, and anteriorly based pegs for additional purchase in cases of poor talar bone stock. While these design elements are geared towards suitability in the rTAA setting, no study to date has reported the efficacy of this component in rTAA. Here, we describe the clinical, radiographic, and survivorship outcomes of patients undergoing rTAA with the use of the talar component of the INVISION ankle replacement system.

## 2. Materials and Methods

### 2.1. Study Design

This was a single-center retrospective study performed at a tertiary academic medical center, of patients who underwent revision TAA using the INVISION Total Ankle Revision System (Wright Medical Group/Stryker, Kalamazoo, MI, USA). After obtaining Institutional Review Board (IRB) approval, the institution’s standardized, prospectively collected TAA registry was queried for all patients who had undergone rTAA. Of 87 total patients who had undergone rTAA, 28 patients received rTAA with the INVISION System and were included in the study. These patients were identified using a manual review of rTAA operative reports. All procedures were performed between August 2018 and September 2022 by foot and ankle trained surgeons with extensive experience performing TAA.

At our institution, selection and use of the INVISION system over other talar dome cut implant options were based on surgeon preference. Decision-making factors typically included insufficient talar bone stock resulting from subsidence of the primary component or large cyst formation. Custom three-dimensional (3D) total talar replacement (TTR) was performed in cases of non-salvageable talar bone loss and were not included in this study. 

### 2.2. Data Variables

The data variables used in this study were obtained through manual chart review. Information about patient demographics and characteristics, such as gender, race, age at primary and rTAA, body mass index (BMI), smoking status, presence of inflammatory arthritis, chronic immunosuppression, and comorbid conditions like depression and anxiety, were documented. Details regarding preoperative medications, including selective serotonin reuptake inhibitors (SSRIs), gabapentin, narcotics, bisphosphonates, calcium/vitamin D supplements, and other osteoporosis medications, were recorded. Laboratory values before surgery, such as hemoglobin, vitamin D levels, c-reactive protein (CRP), erythrocyte sedimentation rate (ESR), white blood count (WBC), WBC aspirate, and hemoglobin A1C, were recorded using manual review of the electronic health record.

Implant information, such as the type of prostheses used in the primary TAA, size of the revision implant components, and whether the revision involved replacing only the talar component or both the talar and tibial components, was documented. The revision implant was also noted.

Secondary procedures performed during the rTAA, such as cotton osteotomy, calcaneal osteotomy, deltoid release, lateral ligament repair, subtalar fusion, talonavicular fusion, gastrocnemius recession, syndesmosis fusion, medial malleolus fixation, and Achilles lengthening, were recorded and analyzed. The total tourniquet time for the rTAA was documented as well.

### 2.3. Radiographic Outcomes 

Anteroposterior (AP) weight-bearing radiographs of the ankle were used to evaluate preoperative, immediate postoperative, and final follow up alignment. Alignment was measured using varus and valgus angles off the tibia and talus. These measurements were evaluated by a foot and ankle surgeon with specialized training. While some patient-to-patient variability existed in this cohort, radiographs were taken at 1 month, 3 months, and 12 months postoperatively.

At preoperative, postoperative, and final follow-up appointments to assess the alignment of the TAA implant components, changes in the alignment of the tibia and talus were measured by comparing preoperative and postoperative radiographs [22]. Alignment changes in terms of varus and valgus were evaluated by a foot and ankle surgeon with specialized training.

### 2.4. Complications

Postoperative complications were identified and diagnosed during subsequent follow-up visits. Common complications included subsequent fractures, nerve injuries, infections, and persistent pain. Minor complications primarily involved symptomatic pain and nerve impingements. Major complications were defined as cases requiring explantation, below-the-knee amputation, or revision of implant components that exceeded a simple polyethylene exchange. Reoperations were defined as additional surgeries related to peri-implant fractures, implant failure, nerve impingement/injury, or infections. 

### 2.5. Statistical Analysis 

The collected data variables were summarized using descriptive statistics. Continuous data were expressed as means (standard deviations) or medians (interquartile ranges, IQR), while categorical variables were presented as counts (percentages). The estimation of survivorship was conducted using Kaplan–Meier analysis. All statistical analyses were conducted using R v4.3.0 (Indianapolis, IN, USA).

## 3. Results

### 3.1. Cohort Characteristics

The characteristics of the cohort are presented in Table 1. Among the 28 revisions using the INVISION revision system, the average age at the primary TAA was 60.68 ± 7.06 years, and at the rTAA, it was 67.07 ± 7.23 years, with 10 male and 18 female patients. The mean BMI of the cohort was 30.84 ± 6.56 kg/m^2^. Nonsmokers accounted for 53.6% (*n* = 15), former smokers for 42.9% (*n* = 12), and there was one active smoker (1.5%). Preoperative diagnoses included rheumatoid arthritis in 14.3% (*n* = 4) and chronic immunosuppression in 17.9% (*n* = 5). Depression was diagnosed in 39.32% (*n* = 11) and anxiety in 21.4% (*n* = 6) preoperatively. SSRIs were actively used by 53.6% (*n* = 15) for anxiety or depression, and 25.0% (*n* = 7) were on gabapentin for reasons such as neuropathic pain. Preoperative narcotic use, including opioids, was reported by 32.1% (*n* = 9) of patients. Preoperative medications affecting bone density were common in the cohort with 67.9% (*n* = 19) using calcium/vitamin D and 10.7% (*n* = 3) using bisphosphonates. Mean follow-up was 1.3 ± 0.9 years.

Table 2 presents the preoperative laboratory values. The mean (SD) hemoglobin was 13.50 (2.03) g/dL, vitamin D level was 49.12 (24.89) ng/mL, CRP was 0.56 (0.30) mg/L, ESR was 19.25 (16.46) mm/hour, WBC was 7.55 (1.69) K/uL, and A1C was 6.03 (0.96) mg/dL. Patients who underwent joint aspiration for suspected infections had a mean WBC aspirate of 280.00 K/uL before their revision surgeries.

The primary TAA utilized various types of prostheses, as shown in Table 1. Vantage (Exactech; Gainesville, FL, USA) accounted for 32.1% (*n* = 9), STAR (DJO; Lewisville, TX, USA) for 32.1% (*n* = 9), INBONE II (Stryker; Kalamazoo, MI, USA) for 10.7% (*n* = 3), INFINITY (Stryker; Kalamazoo, MI, USA) for 10.7% (*n* = 3), Agility (Depuy, Warsaw, IN, USA) for 3.6% (*n* = 1), INBONE I (Stryker: Kalamazoo, MI, USA) for 3.6% (*n* = 1), Salto Talaris Anatomic Ankle (Integra LifeSciences, Princeton, NJ, USA) for 3.6% (*n* = 1), and Zimmer Trabecular Metal (Zimmer Biomet, Warsaw, IN, USA) for 3.6% (1 case). In only one case (3.6%), the talar component of the implant was exchanged while the tibial component remained unchanged. All 28 patients undergoing revision TAA with the INVISION system received a stemmed protheses as their revision implant.

Secondary procedures were performed in 89.3% (*n* = 25) of patients during the rTAA while using the INVISION system (Table 3). Among these additional procedures, the most prevalent were medial malleolus fixation (78.6%, *n* = 22), gastrocnemius recession (50.0%, *n* = 14), deltoid release (28.6%, *n* = 8), lateral ligament repair with suture anchor (10.7%, *n* = 3), subtalar fusion (10.7%, *n* = 3), talonavicular fusion (3.6%, *n* = 1), and cotton osteotomy (3.6%, *n* = 1). The mean duration of tourniquet application was 2.75 ± 0.53 h.

### 3.2. Radiographic Outcomes 

Radiographic outcomes were assessed and are presented in Table 4, documenting alignment measurements at the preoperative, postoperative, and final follow-up stages. Prior to surgery, the tibia varus and valgus alignment measurements were 4.07 (3.06) degrees and 3.67 (4.36) degrees, respectively, while the talus varus and valgus alignment measurements were 4.83 (6.68) degrees and 4.22 (4.21) degrees. Immediate postoperative radiographs indicated alignment measurements of 1.75 (2.44) degrees and 2.40 (3.75) degrees for tibia varus and valgus, respectively, and 1.08 (2.18) degrees and 2.68 (4.21) degrees for talus varus and valgus, respectively. At the final postoperative follow-up, the alignment measurements were 1.67 (2.16) degrees and 2.00 (3.48) degrees for tibia varus and valgus, and 1.23 (2.45) degrees and 2.32 (3.93) degrees for talus varus and valgus, respectively. Weightbearing anterior-posterior and lateral radiographs of a failed and successful rTAA using the INVISION system can be seen in Figure 1.

### 3.3. Complications 

Complications were reported by 39.3% (*n* = 11) of those using the INVISION Total Ankle Revision System, most commonly implant failure (*n* = 3), infection (*n* = 3), poor wound healing (*n* = 2), and nerve injury/impingement (*n* = 2). Minor complications were observed in 25.0% (*n* = 7) of the cohort. At 3-year follow-up, 45.8% (23.7–88.6) of patients remained free from any diagnosed or reported complications (Figure 2). Details of complications are presented in Table 5.

Major complications were observed in 14.3% (*n* = 4) of the INVISION cohort (Figure 2). The most frequently observed major complication was infection (*n* = 3). One patient on immunomodulatory medications for rheumatoid arthritis suffered a traumatic foot injury ultimately leading to a prosthetic joint infection with wound dehiscence at the revision site. The infection did not resolve with antibiotic treatment and the patient underwent BKA. Another patient developed septic arthritis of the revised ankle and underwent irrigation and debridement and exchange of both the polyethylene and talar component. A third patient presented with significant pain and drainage from their incision at six-week follow up. Aspiration cultures grew *Enterococcus faecalis* and the patient was admitted for incision and drainage, explanation of revision implant, and placement of an antibiotic spacer. A single patient experienced mechanical implant failure. The patient presented asymptomatically for routine six-week follow up and was found to have a distal tibia fracture with malposition of the of the revision implant showing a shift in the tibial stem and tibia baseplate disengagement. Tibiotalocalcaneal fusion was discussed, though the patient ultimately decided not to undergo reoperation. Overall, 10.7% (*n* = 3) of patients underwent reoperation.

## 4. Discussion

The growing incidence of TAA as a treatment for end-stage ankle arthritis indicates that necessity for rTAA will increase in coming years. Thus, a close examination of revision surgery outcomes is prudent. While modern ankle prostheses have improved implant survival rates, primary TAA replacement continues to have higher failure rates compared to hip and knee arthroplasty [23,24]. Thus, rTAA is likely to be performed at higher rates than revision total hip and total knee arthroplasty [5]. Despite encouraging evidence that revision ankle replacement can enhance functionality, it carries notable risks of failure and subsequent surgeries, especially when compared to primary procedures [25,26,27,28]. Thus, further study of revision arthroplasty is imperative to improving patient outcomes.

The INVISION system (Wright Medical/Stryker in Kalamazoo, MI, USA) is a notable advancement in ankle replacement technology, particularly for revision arthroplasty. Its modular design facilitates compatibility with both the INBONE II talar and tibial components, as well as the INFINITY tibial components. Notable adaptations for revision procedures include a larger tibial tray and an extended anterior talar component that provides enhanced support for the implant on robust cortical bone structures. Due to the novelty of the device as a revision-specific system, there is little published information on the INVISION system across the literature. To our knowledge, this is the first publication to evaluate outcomes of revision TAA using the INVISION talar component.

Overall complication rates following rTAA (39.3%) with a mean follow up of 1.3 ± 0.9 years were higher in our cohort than reported by other studies [26]. We provide a thorough account of all complications possibly linked to the revision procedure including persistent pain, nerve injury, and neuropathies that may have been related to the rTAA and non-union of concomitant arthrodesis procedures. Discrepancies in the system registries used to classify complications may explain our elevated complication rate, many of which were deemed “minor complications” upon review.

Despite the relatively low rate of survival free from both minor and major complications (45.8%), our investigation revealed that both the overall reoperation rate (10.7%) and the incidence of implant failure (10.7%) were comparatively lower than the rates reported in the existing literature regarding rTAA [25,26,28]. A recent systematic review conducted by Jennison et al. evaluated rTAA outcomes across 15 publications and 397 patients [29]. The authors found the overall need for further surgical intervention was 26.9%, and the pooled percentage of procedures that required re-revision for failure was 14.4%. The mean follow-up time up was a range of 1–6.9 years, making direct comparisons with our cohort challenging. Nevertheless, our results demonstrate strong short-term durability of the INVISION.

Complication and re-operation rates of rTAA vary widely across the literature [28,29,30,31,32,33]. Kamrad et al. demonstrated in a retrospective case series of 61 patients undergoing rTAA that 40.5% (*n* = 28) of patients underwent re-operation of their primary revision and that 34% (*n* = 21) of the primary revisions failed at a median of 26 months. This amounted to a total of 47 additional surgical procedures, 72% of which were an additional revision of the index rTAA procedure [31]. Alternatively, in a retrospective case series of 29 patients undergoing rTAA with a mean follow of 3.2 years post-revision, the authors reported that 10.3% (*n* = 3) of patients required further surgery [30]. This variation in re-operation and implant failure rates across studies may be attributed to a number of factors, including variability in follow-up duration, small sample size, differing definitions of outcomes and revision procedures (i.e., isolated polyethylene exchange versus two-component exchange arthroplasty), and study inclusion and exclusion criteria. Further research and meta-analyses may help provide a clearer understanding of the true success rates of rTAA and the factors contributing to the variability observed.

In our study, 75% (*n* = 3) of major complications and 100% of reoperations (*n* = 3) were due to infection of the prostheses. The remaining 25% (*n* = 1) were caused by mechanical failure of the implant. In this case, routine radiographs at six-week follow-up revealed distal tibial fracture with malposition of the rTAA implant on the tibial side, showing a shift of the tibial stem and disengagement of the baseplate. The patient elected not to undergo re-operation. Other forms of implant failure were deemed as minor complications and accounted for 28.6% (*n* = 2) of minor complications. One patient experienced slight interval collapse of the talar component of the implant, which was noted on postoperative radiographic imaging. The other patient had interval subsidence of the talar component, with progressive hindfoot valgus and associated pes planus deformity. Neither of these patients underwent re-operation given the lack of substantial implant-related symptomology. Given the complication rate observed in this study and the rates reported more broadly in the literature, it is important to note that such rates are generally not acceptable for orthopedic procedures. This underscores the need for continued advancement in TAA techniques and implant designs, as well as better delineation of risk factors for complications and methods to mitigate these risks.

In summation, our results indicate that major complications and reoperations primarily stemmed from prosthetic infection rather than from implant failure. This outcome underscores the potential safety and reliability of the INVISION system for rTAA in our cohort. Long-term follow-up studies are warranted to provide a more comprehensive assessment of the implant’s performance and durability.

Restoring neutral ankle alignment at the time of rTAA has been shown to improve gait mechanics at 24-month follow-up and can influence the longevity of implant components. Thus, the sustained improvements in tibial and talar varus and valgus postoperative alignment further support the efficacy of the INVISION system. While the statistical significance of these improvements was limited by the size of our cohort, when considered alongside the low rates of failure and re-operation, these numbers suggest that the INVISION system could offer a durable option for revision arthroplasty with regards to coronal plane alignment.

In addition to the mechanical benefits conferred by the INVISION system in maintaining postoperative alignment, the comprehensive rehabilitation process, including physical therapy (PT), emerges as a pivotal factor in maximizing patient outcomes [34,35]. While the role of postoperative rehabilitation is well established in hip and knee arthroplasty, there is scarce literature evaluating its impact after TAA [36,37,38,39,40,41]. Furthermore, though the positive impacts of TAA and rTAA on range of motion and pain have led to its rising popularity, these procedures do not restore gait mechanics to levels experienced by age-matched persons without arthritis [42]. Thus, it becomes imperative to examine the role of postoperative PT in further enhancing these functional outcomes. Physical therapy, tailored to address specific defecits in strength and proprioception, may both expedite recovery and potentiate the long-term benefits of improved coronal plane alignment. Future studies with larger cohorts are warranted to quantify the impact of PT regimens on functional recovery and to establish evidence-based protocols that can be standardized across revision ankle arthroplasty procedures.

Our study had several limitations. Despite the comparably strong cohort size of 28 patients receiving the INVISION system for rTAA, the small sample size relative to studies evaluating primary TAA and the retrospective nature of the study made conducting statistical analysis challenging, particularly when controlling for various types of primary implants and concomitant procedures.

Additionally, our study did not capture patient reported outcomes (PROs) and instead focused on survivorship from complications and radiographic alignment. Evaluation of functional recovery including pain and range of motion is paramount to understanding the efficacy of implant designs, and future evaluations of the INVISION system must aim to include these elements to better understand the implants’ efficacy. However, given the current use of the INVISION system, early term safety is an important consideration that should be comprehensively reported to establish a baseline for long-term monitoring and follow up.

The absence of PRO metrics, including PROMIS scores, in our study may affect the interpretation of surgical outcomes by not reflecting the patient’s perception of recovery and quality of life post-surgery. These limitations can be addressed in future studies by including a broader array of outcome measures that might provide a more holistic view of the patient’s postoperative journey. Such prospective studies should strive to expand the cohort size and to methodically capture PROs, thus validating the efficacy of the INVISION system. Despite these limitations, we believe that this early series provides a necessary contribution to the literature regarding the safety and early-term outcomes of revision-specific implants.

## 5. Conclusions

Our study offers early insight into the performance of the INVISION system, which notably improved rates of re-operation and implant failure. While overall complication rates were high (39.3%, *n* = 11), the majority of these were attributed to minor complications (25%, *n* = 7). The rate of major complications (10.7%, *n* = 3) and revisions (10.7%, *n* = 3) were relatively lower than rates reported in the literature. Moreover, analysis of radiographic changes in varus and valgus alignment showed durable improvements at final follow-up. Despite the limitations inherent in this initial investigation, we present promising findings regarding the effectiveness of utilizing revision-specific implant systems. As the adoption of this pivotal innovation in ankle arthroplasty continues to gain traction, we anticipate opportunities for larger studies with more robust outcome metrics. These early findings lay a foundation for improving the management of the failed TAA.

## Figures and Tables

**Figure 1 diagnostics-14-01612-f001:**
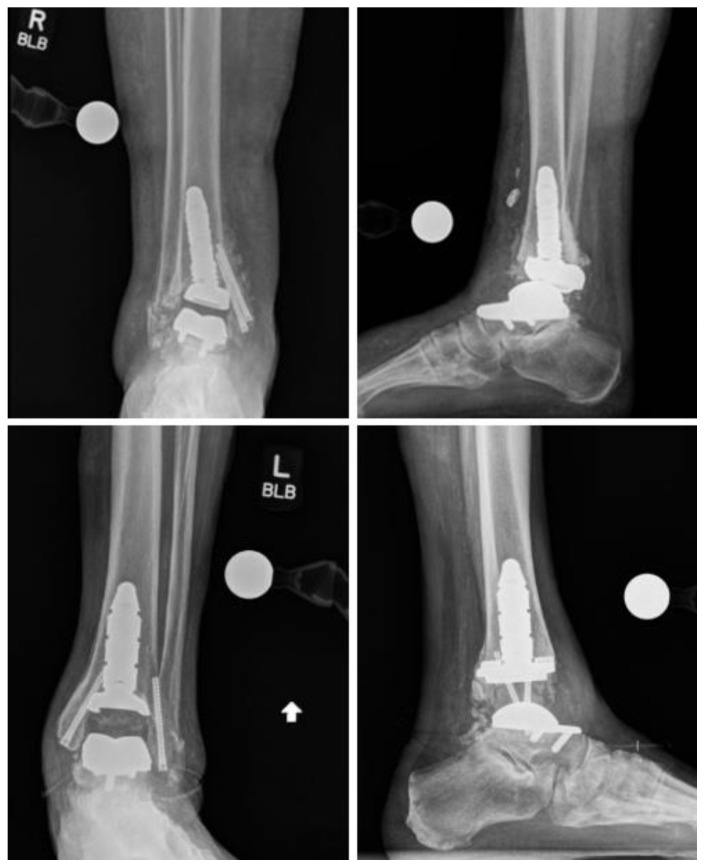
Failed (**top**) and successful (**bottom**) AP and lateral radiographs of weightbearing total ankle arthroplasty using the INVISION system.

**Figure 2 diagnostics-14-01612-f002:**
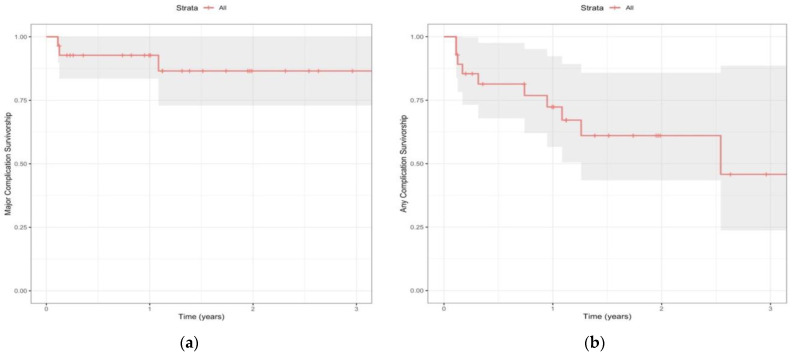
The Kaplan–Meier curves above present the survival analysis of (**a**) all complications and (**b**) major complications in patients undergoing revision surgery using the INVISION Total Ankle Revision System. The *x*-axis represents the follow-up time in years, while the *y*-axis represents the proportion of patients who remained free from complications.

**Table 1 diagnostics-14-01612-t001:** Patient characteristics and preoperative implant type.

Factor	Study Group (*n* = 28)
Gender, *n*	18 Female, 10 Male
Age at primary, mean (sd)	60.68 (7.06)
Age at revision, mean (sd)	67.07 (7.23)
BMI, mean (sd)	30.84 (6.56)
Smoking status, *n* (%)	
Current smoker	1 (3.6)
Never a smoker	15 (53.6)
Previous smoker	12 (42.9)
Rheumatoid arthritis, *n* (%)	4 (14.3)
Chronic immunosuppression, *n* (%)	5 (17.9)
Preoperative depression, *n* (%)	11 (39.3)
Preoperative anxiety, *n* (%)	6 (21.4)
Preoperative SSRI use, *n* (%)	15 (53.6)
Preoperative gabapentin use, *n* (%)	7 (25.0)
Preoperative narcotic use, *n* (%)	9 (32.1)
Preoperative bisphosphonate use, *n* (%)	3 (10.7)
Preoperative other osteoporosis medication use, *n* (%)	28 (100.0)
Preoperative calcium/vitamin D use, *n* (%)	19 (67.9)
Preoperative implant, *n* (%)	
Agility	1 (3.6)
InBone I	1 (3.6)
InBone II	3 (10.7)
INFINITY	3 (10.7)
Salto Talaris	1 (3.6)
STAR	9 (32.1)
Vantage	9 (32.1)
ZTM	1 (3.6)

Abbreviations: *n*, number; sd, standard deviation; SSRI, selective serotonin reuptake inhibitor; ZTM, Zimmer Trabecular Metal.

**Table 2 diagnostics-14-01612-t002:** Preoperative laboratory values.

Laboratory Test	Value
Preoperative Hgb, mean (sd)	13.50 (2.03)
Preoperative vitamin D, mean (sd)	49.12 (24.89)
Preoperative CRP, mean (sd)	0.56 (0.30)
Preoperative ESR, mean (sd)	19.25 (16.46)
Preoperative WBC, mean (sd)	7.55 (1.69)
Preoperative WBC Aspirate, mean	280
Preoperative A1c, mean (sd)	6.03 (0.96)

Abbreviations: Hgb, hemoglobin; sd, standard deviation; CRP, c-reactive protein; ESR, erythrocyte sedimentation rate; WBC, white blood count; A1c, hemoglobin A1C.

**Table 3 diagnostics-14-01612-t003:** Secondary procedures performed with rTAA using the INVISION system.

Procedures	Study Group (*n* = 28)
Deltoid release, *n* (%)	8 (28.6)
Lateral ligament repair with suture anchor, *n* (%)	3 (10.7)
Medial malleolus fixation, *n* (%)	22 (78.6)
Subtalar fusion, *n* (%)	3 (10.7)
Talonavicular fusion, *n* (%)	1 (3.6)
Cotton osteotomy, *n* (%)	1 (3.6)
Gastrocnemius recession, *n* (%)	14 (50.0)
Tourniquet time (h), mean (sd)	2.75 (0.53)

Abbreviations: *n*, number; sd, standard deviation; h, hours.

**Table 4 diagnostics-14-01612-t004:** Radiographic outcomes of patients undergoing rTAA using the INVISION system, including preoperative, immediate postoperative, and final postoperative varus and valgus measurements of the talus and tibia.

Radiographic Measures	Preoperative Radiograph	Immediate Postoperative Radiograph	Final Preoperative Radiograph
Tibia deg, mean (sd)			
Varus	4.07 (3.06)	1.75 (2.44)	1.67 (2.16)
Valgus	3.67 (4.36)	2.40 (3.75)	2.00 (3.48)
Talus deg, mean (sd)			
Varus	4.83 (6.68)	1.08 (2.18)	1.23 (2.45)
Valgus	4.22 (4.21)	2.68 (4.21)	2.32 (3.93)

Abbreviations: deg, degrees; sd, standard deviation.

**Table 5 diagnostics-14-01612-t005:** Complications and reoperations following rTAA using the INVISION system.

Outcome	Total, *n* (%)
Any complication	11 (39.3)
Major complication	4 (14.3)
Infection	3 (10.7)
Asymptomatic peri-implant lucency	1 (3.4)
Minor complication	7 (25.0)
Tibial nerve injury	2 (7.1)
Impingement	1 (3.4)
Poor wound healing	1 (3.4)
Subtalar joint non-union	1 (3.4)
Interval collapse (talar component)	1 (3.4)
Subsidence (talar component)	1 (3.4)
Revision	3 (10.7)
Total ankle arthroplasty	1 (3.4)
Below-the-knee amputation	1 (3.4)
Implantation of an antibiotic spacer	1 (3.4)

## Data Availability

The data that support the findings of this study are not publicly available due to privacy and ethical restrictions. The data contain personally identifiable information or confidential patient records, which is protected under privacy laws and regulations. Access to the data is therefore restricted and not openly disclosed. Further inquiries can be directed to the corresponding author, who will process the requests on a case-by-case basis.

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
