# Peer review of "The INVISION Talar Component in Revision Total Ankle Arthroplasty: Analysis of Early Outcomes"

_diagnostics, 2024, doi:10.3390/diagnostics14151612_

Round 1

Reviewer 1 Report

Comments and Suggestions for Authors

This article analyzes the results of the INVISION Talar Component in Revision Total Ankle Ar- 2 throplasty. The study appears to have been conducted in an exhaustive and very detailed manner, compliments to all the authors. However, there are some things I would recommend reviewing. Below are suggestions for reviewing. - The results and conclusions in the abstract appear a bit mediocre. It would be useful to provide some more statistical data. I would also divide the abstract into the classic points: Introduction, materials and methods, etc. - Line 32: It would be useful to mention some more recent works such as: Li T, Zhao L, Liu Y, Huang L, Zhu J, Xiong J, Pang J, Qin L, Huang Z, Xu Y, Dai H. Total ankle replacement versus ankle fusion for end-stage ankle arthritis: A meta- analysis. J Orthop Surg (Hong Kong). 2024 Jan-Apr;32(1):10225536241244825. doi: 10.1177/10225536241244825. PMID: 38607239. Samaila EM, Bissoli A, Argentini E, Negri S, Colò G, Magnan B. Total ankle replacement in young patients. Acta Biomed. 2020 May 30;91(4-S):31-35. doi: 10.23750/abm.v91i4-S.9725. PMID: 32555074; PMCID: PMC7944830. - Line 73: the Mean follow-up should be put in the results. - Line 97: "Concurrent procedures" should be replaced with "secondary procedures" - Line 102: How was hindfoot alignment evaluated? with the Saltzman view? please specify. - Line 103: At how many months were the patients re-evaluated using new x-rays? 3-6-12 months? Please specify. - Line 262: Discussions should be put together, without subparagraphs. - Conclusions: they appear very poor in data. Please integrate them. In general the references appear a little too scarce, it would be useful to cite at least 5-10 more studies.

Author Response

Reviewer 1:

This article analyzes the results of the INVISION Talar Component in Revision Total Ankle Ar- 2 throplasty. The study appears to have been conducted in an exhaustive and very detailed manner, compliments to all the authors. However, there are some things I would recommend reviewing. Below are suggestions for reviewing. - The results and conclusions in the abstract appear a bit mediocre. It would be useful to provide some more statistical data. I would also divide the abstract into the classic points: Introduction, materials and methods, etc. - Line 32: It would be useful to mention some more recent works such as: Li T, Zhao L, Liu Y, Huang L, Zhu J, Xiong J, Pang J, Qin L, Huang Z, Xu Y, Dai H. Total ankle replacement versus ankle fusion for end-stage ankle arthritis: A meta- analysis. J Orthop Surg (Hong Kong). 2024 Jan-Apr;32(1):10225536241244825. doi: 10.1177/10225536241244825. PMID: 38607239. Samaila EM, Bissoli A, Argentini E, Negri S, Colò G, Magnan B. Total ankle replacement in young patients. Acta Biomed. 2020 May 30;91(4-S):31-35. doi: 10.23750/abm.v91i4-S.9725. PMID: 32555074; PMCID: PMC7944830. - Line 73: the Mean follow-up should be put in the results. - Line 97: "Concurrent procedures" should be replaced with "secondary procedures" - Line 102: How was hindfoot alignment evaluated? with the Saltzman view? please specify. - Line 103: At how many months were the patients re-evaluated using new x-rays? 3-6-12 months? Please specify. - Line 262: Discussions should be put together, without subparagraphs. - Conclusions: they appear very poor in data. Please integrate them. In general the references appear a little too scarce, it would be useful to cite at least 5-10 more studies.

Comments 1: The results and conclusions in the abstract appear a bit mediocre. It would be useful to provide some more statistical data. I would also divide the abstract into the classic points: Introduction, materials and methods, etc.

Response 1: Thank you for pointing this out. We agree with your comment and have made the following changes to the abstract. We have broken it up into the sections mentioned (lines 12-36, page 1) and included additional statistics in the results and conclusions (lines 20-22 and lines 26-29, page 1).

Comments 2: Line 32: It would be useful to mention some more recent works such as: Li T, Zhao L, Liu Y, Huang L, Zhu J, Xiong J, Pang J, Qin L, Huang Z, Xu Y, Dai H. Total ankle replacement versus ankle fusion for end-stage ankle arthritis: A meta- analysis. J Orthop Surg (Hong Kong). 2024 Jan-Apr;32(1):10225536241244825. doi: 10.1177/10225536241244825. PMID: 38607239. Samaila EM, Bissoli A, Argentini E, Negri S, Colò G, Magnan B. Total ankle replacement in young patients. Acta Biomed. 2020 May 30;91(4-S):31-35. doi: 10.23750/abm.v91i4-S.9725. PMID: 32555074; PMCID: PMC7944830.

Response 2: Thank you for pointing this out. We agree with your comment and have made the following changes. We have cited and discussed the requested studies in the introduction (lines 53-72, page 2)

Comments 3: Line 73: the Mean follow-up should be put in the results.

Response 3: We thank you for pointing this out and have moved the mean follow up to the results (results, page 4, lines 178)

Comment 4: Line 97: "Concurrent procedures" should be replaced with "secondary procedures"

Response 4: We thank you for pointing this out. We have changed concurrent procedures to secondary procedures at all points throughout the manuscript

Comment 5: Line 102: How was hindfoot alignment evaluated? with the Saltzman view? please specify.

Response 5: We thank you for pointing this out. We clarified the views used to evaluate alignment (AP radiograph of the ankle) and tried to better explain these measurements. (Methods, page 3, lines 133-138)

Comment 6: Line 103: At how many months were the patients re-evaluated using new x-rays? 3-6-12 months? Please specify.

Response 6: We thank you for pointing this out. Although the x-rays were not all collected at the exact same time points given variability in patient follow up, we have specified the routine post operative x-ray protocol for our total ankle arthroplasty patients. (methods, page 3, lines 136-138)

Comment 7: Line 262: Discussions should be put together, without subparagraphs.

Response 7: We thank you for pointing this out and agree with your comment. We have taken out the sub paragraphs for the discussion (see updated manuscript).

Comment 8: Conclusions: they appear very poor in data. Please integrate them.

Response 8: We thank you for pointing this out and agree with your comment. We have added additional data to the conclusion (conclusion, page 11, lines 438-442).

Comment 9: In general the references appear a little too scarce, it would be useful to cite at least 5-10 more studies.

Response 9: We agree with your comment and have updated our references including more recent references mentioned previously in comment 2. Please see introduction page 2, lines 53-72.

Reviewer 2 Report

Comments and Suggestions for Authors

Well written report on the early results of a revision ankle arthroplasty system by Wright Stryker. It concerns a restrospective cohort of limited size that reports on the most important objective outcome parameters, however does not report PROMS. The retrospective nature of the study inherently poses issues of selection bias and information bias. The extent of the latter could be minimal if information drawn from the medical charts concerns information that is consistently collected from all patients (i.e. standardized registry that collects the mentioned base line characteristics and outcome measures in a constant manner). Was this the case in this study?

Moreover, the authors are quite content with the results as can be imagined in these highly complex cases in which an earlier TAA has failed. Hence, the high complication is seen by the authors in this context and compared to literature that also shows high complication rates (albeit slightly less high). I do think that in the discussion, a sentence or two concerning the consequences of this high complication rate (a complication rate like this is not accepted in general for (orthopaedic) procedures), is warranted. 

When these issues are addressed, the article in my opinion does indeed bring valuable new data to the field.

Comments on the Quality of English Language

English needs some minor editing.

Author Response

Reviewer 2:

Well written report on the early results of a revision ankle arthroplasty system by Wright Stryker. It concerns a restrospective cohort of limited size that reports on the most important objective outcome parameters, however does not report PROMS. The retrospective nature of the study inherently poses issues of selection bias and information bias. The extent of the latter could be minimal if information drawn from the medical charts concerns information that is consistently collected from all patients (i.e. standardized registry that collects the mentioned base line characteristics and outcome measures in a constant manner). Was this the case in this study?

Moreover, the authors are quite content with the results as can be imagined in these highly complex cases in which an earlier TAA has failed. Hence, the high complication is seen by the authors in this context and compared to literature that also shows high complication rates (albeit slightly less high). I do think that in the discussion, a sentence or two concerning the consequences of this high complication rate (a complication rate like this is not accepted in general for (orthopaedic) procedures), is warranted.

When these issues are addressed, the article in my opinion does indeed bring valuable new data to the field.

Comment 1: The extent of the latter could be minimal if information drawn from the medical charts concerns information that is consistently collected from all patients (i.e. standardized registry that collects the mentioned base line characteristics and outcome measures in a constant manner). Was this the case in this study?

Response 1: We thank you for this comment and agree with you. This review was completed using a data from a routinely collected data base. We have specified this in the methods, page 3, lines 98-99.

Comment 2: I do think that in the discussion, a sentence or two concerning the consequences of this high complication rate (a complication rate like this is not accepted in general for (orthopaedic) procedures), is warranted.

Response 2: We agree with your comment and have added the requested context to the discussion. Please see the discussion, page 10, lines  379-383.

We thank you for your comments. All changes are included in red with track changes on.